# Downregulation of Ribosomal Contents and Kinase Activities Is Associated with the Inhibitive Effect on the Growth of Group B Streptococcus Induced by Placental Extracellular Vesicles

**DOI:** 10.3390/biology10070664

**Published:** 2021-07-14

**Authors:** Jing Gao, Yunhui Tang, Xinyi Sun, Qiujing Chen, Yiqian Peng, Catherine Jia-Yun Tsai, Qi Chen

**Affiliations:** 1Department of Medical Laboratory, The Hospital of Obstetrics & Gynaecology, Fudan University, Shanghai 200081, China; gaojing1511@163.com (J.G.); pengyiqian6298@fckyy.org.cn (Y.P.); 2Department of Family Planning, The Hospital of Obstetrics & Gynaecology, Fudan University, Shanghai 200081, China; 3Department of Obstetrics & Gynaecology, The University of Auckland, Auckland 1142, New Zealand; xinyi_sun_970623@outlook.com (X.S.); q.chen@auckland.ac.nz (Q.C.); 4Institute of Cardiovascular Disease, Ruijing Hospital, School of Medicine, Shanghai Jiaotong University, Shanghai 200081, China; 13564670350@163.com; 5Department of Molecular Medicine and Pathology, The University of Auckland, Auckland 1142, New Zealand; j.tsai@auckland.ac.nz

**Keywords:** placental EVs, GBS, susceptibility, proteomics, protein synthesis, cell energy

## Abstract

**Simple Summary:**

The bioactive properties of extracellular vesicles (EVs) in the physiological and pathophysiological conditions of pregnancy and cancers have been well investigated. However, the interaction of EVs and bacterial cells has not been studied yet. Transferring and releasing of cargos in EVs intracellularly impact the function of target cells. In this study, we investigated the effect of placental EVs on the growth of opportunistic pathogens such as Group B Streptococcus (GBS), which is a major cause of some complications of pregnancy. We found that placental micro-EVs or nano-EVs attenuated the growth of a Gram-positive bacterium, GBS. This attenuative effect at least partially required an interaction of placental EVs with GBS. Proteomic analysis showed that changes in protein synthesis or cellular energy in GBS may contribute to this inhibitory effect.

**Abstract:**

Background: Like many other cell types, the human placenta produces large amounts of extracellular vesicles (EVs). Increasing evidence has shown that placental EVs contribute to the regulation of maternal immune and vascular systems during pregnancy via the transfer of their cargos. In this study, we investigated the effect of placental EVs on the growth of opportunistic pathogens that commonly colonise the female reproductive tract. Methods: Gram-positive bacterium Group B Streptococcus (GBS) and Gram-negative bacterium Escherichia coli (*E. coli*) were treated with placental EVs that were collected from placental explant cultures, and the growth, susceptibility, and resistance to antibiotics of the bacteria were measured. In addition, comparative proteomics analysis was also performed for the GBS with or without exposure to placental EVs. Results: When treated with placental micro-EVs or nano-EVs, the GBS growth curve entered the stationary phase earlier, compared to untreated GBS. Treatment with placental EVs also inhibited the growth of GBS on solid medium, compared to untreated GBS. However, these biological activities were not seen in *E. coli*. This attenuative effect required interaction of placental EVs with GBS but not phagocytosis. In addition, the susceptibility or resistance to antibiotics of GBS or *E. coli* was not directly affected by treatment with placental EVs. The proteomic and Western blotting analysis of GBS that had been treated with placental EVs suggested that the downregulation of cellular components and proteins associated with phosphorylation and cell energy in GBS may contribute to these attenuative effects. Conclusion: We demonstrated the attenuative effect of the growth of GBS treated with placental EVs. Downregulation of cellular components and proteins associated with phosphorylation and cell energy may contribute to the physiological changes in GBS treated with placental EVs.

## 1. Introduction

Group B Streptococcus (GBS) is a Gram-positive bacterium often found in the urinary tract, digestive system, and reproductive tract. It is a normal commensal in about 25% of all healthy adult women. GBS can also be found in pregnant women’s vaginal or urinary tract, or placenta or womb and amniotic fluid (reviewed in [1]). GBS can be passed to infants during the process of labour and delivery [2,3]. In addition, GBS is a major cause of preterm birth and stillbirth in some serious cases [4,5]. Although pregnant women with GBS infection can be treated with a number of antibiotics before delivery, GBS remains an obstetrical problem in clinical practice. 

Extracellular vesicles (EVs) are lipid bilayer-enclosed packages of cellular contents that are involved in cell–cell, cell–organ, and cell–organism communication and signalling. These vesicles are extruded from various cell types and carry cargos of proteins, regulatory RNAs, DNA, and lipids which reportedly could be biologically active. During pregnancy, a large number of cellular vesicles are extruded from a single multinucleated cell (syncytiotrophoblast) which is located in the surface layer of the human placenta, into the maternal blood, as early as after 6 weeks of pregnancy [6,7]. These placental EVs can be divided into three subtypes by size, namely macro-EVs (also called multinucleated syncytial nuclear aggregates (SNAs)), micro-EVs, and nano-EVs (including exosomes) (reviewed in [8]). These lipid bilayer-enclosed placental EVs contain cargos of proteins, DNA, RNA, and lipid. There is increasing evidence suggesting that placental EVs are involved in maternal physiological adaptation during pregnancy [9], and placental EVs also contribute to the pathogenesis of complications of pregnancy, such as preeclampsia [10,11] and gestational diabetes mellitus (GDM) [12].

A number of studies have indicated that placental EVs contribute to the regulation of maternal immune and vascular systems during pregnancy [13,14,15,16,17,18]. We have previously reported that phagocytosis of placental EVs from normal pregnancies makes endothelial cells resistant to activation in vitro [17]. In addition, we recently reported that placental EVs, but not EVs from other sources, inhibited proliferation and cell cycle progression of ovarian cancer cells following phagocytosis of the EVs [19]. Although the underlying mechanisms of these modulatory activities are slowly emerging, our studies have shown that some of these effects are mediated by the proteins from placental EVs [20], whilst other effects are mediated by regulatory RNAs transferred via placental EVs to maternal cells [21]. These studies suggested that placental EVs could affect the functions of target cells. We recently identified 1952 proteins in placental EVs, many of which are involved in controlling cell growth, by a non-quantitative analysis [20].

To date, there has been little information on the interaction of EVs and bacteria. Since placental EVs play multiple roles in pregnancy, such as affecting the function of target cells, in this study we aimed to investigate the effect of placental EVs on the growth of opportunistic pathogens such as GBS. We hypothesised that placental EVs possess antimicrobial properties that impede bacterial growth, and therefore provide protection against GBS infections.

## 2. Methods

This study received approval by the Ethics Committee of the Hospital of Obstetrics & Gynaecology, Fudan University, China (reference number: 2020-11). This study conforms to the principles outlined in the Declaration of Helsinki. All patient-derived tissues were obtained following informed written consent.

### 2.1. Collection of Placental EVs

Placental EVs (micro- and nano-EVs) were collected from first trimester placentae (*n* = 8) from elective surgical abortion, as we previously described [22]. Briefly, approximately 400 mg placental explants were dissected and cultured in Netwell™ culture inserts (with 440 µm mesh size) in 12-well culture plates for 18 h. In some experiments, placental explants were labelled with Cell Tracker Red (CMTPX, Invitrogen, Shanghai, China) (1 µM) for 18 h as we previously described [19]. The conditioned media were then collected and centrifuged at 2000× *g* for 5 min to remove cellular debris. The supernatant was then centrifuged at 20,000× *g* for 1 h for the collection of micro-EVs. The remaining supernatant after micro-EV collection was further centrifuged at 100,000× *g* for 1 h for the collection of nano-EVs. Isolated placental micro-EVs and nano-EVs were suspended in PBS and stored at 4 °C and used within 3 days. The concentration of placental EVs was measured by s bicinchoninic acid (BCA) protein assay following the manufacturer’s instructions (ThermoFisher Scientific, Shanghai, China). The characterisations of placental EVs were confirmed by morphology (electronic microscopy) and CD 81 expression (data not shown).

### 2.2. Collection of Bacteria 

Gram-positive bacteria, Group B Streptococcus (GBS, *Strepcococcus agalactiae*), or Gram-negative bacteria (*Escherichia coli* (*E. coli*)) were collected from positive blood culture samples from pregnant women with bacterial blood stream infections from our hospital’s medical laboratory. All samples were collected from at least three different patients for biological replication.

### 2.3. Measurement of Bacterial Growth

A single colony of GBS or *E. coli* was picked from culture plate, and then cultured in 5 mL brain–heart infusion (BHI) or lysogeny broth (LB), respectively, overnight at 37 °C. On the next day, 50 µL of the culture were then added to 6 mL of fresh media in the presence or absence of 500 µg (total proteins) micro- or nano-EVs and incubated at 37 °C. The optical density (OD) value was measured at each 30 min point at 600 nm on an Eppendorf BioPhotometer^®^ D30 (Shanghai, China). In other experiments, 50 µL of the prepared culture (described above) were added to 6 mL of fresh media in the presence or absence of 500 µg (total proteins) micro- or nano-EVs, and cultured for 0, 3, 6, and 12 h. At each time point, a sample from the culture was taken and serially diluted. Then, 10 µL of the 10^−4^ dilution was drop-plated on Columbia blood agar plates, and cultured for 24 h at 37 °C for enumeration. In some experiments, cytochalasin D (an inhibitor of phagocytosis, 10 µM) was added into the cultures. All experiments were repeated at least three times.

### 2.4. Determination of the Interaction of Placental EVs with Bacteria 

A single colony of GBS or *E. coli* was used to inoculate 5 mL brain–heart infusion (BHI) or lysogeny broth (LB), respectively, overnight at 37 °C. On the next day, 15 µL of the culture were then added to 1.5 mL of fresh media in the presence or absence of 150 µg (total proteins) red fluorescently labelled (CMTPX) micro- or nano-EVs and incubated at 37 °C for 6 h. After washing with PBS and centrifugation, the pellets of bacteria were suspended in PBS and the fluorescent intensity was measured by a fluorescent plate reader at 520/630 nm (Synergie 2, BioTek, Winooski, Vermont, U.S.A). In some experiments, cytochalasin D (an inhibitor of phagocytosis, 10 µM) was added into the cultures.

### 2.5. Measurement of Antibiotic Susceptibility

The susceptibility or resistance of bacteria to antibiotics was measured using the Kirby–Bauer method [23]. Briefly, GBS or *E. coli* suspended in 3 mL of 0.5 McDonald’s standard in the presence or absence of placental micro-EVs (300 µg) or nano-EVs (300 µg) were lawn-plated on Mueller–Hinton agar plates supplemented with 5% sheep serum. Small paper discs containing antibiotics (ampicillin 10 µg; penicillin 15 units; erythromycin 15 µg; linezolid 30 µg; vancomycin 30 µg and clindamycin 2 µg for GBS, or fosfomycin 200 µg; piperacillin/tazobactam 100/10 µg; meropenem 10 µg; ceftazidime 30 µg; ciprofloxacin 5 µg; amikacin 30 µg; ampicillin 10 µg; piperacillin 100 μg; ampicillin/sulbactam 10/10 µg; ceftriaxone 30 µg for *E. coli*) were then placed onto the plates. Alternatively, in some experiments, small paper discs containing placental micro-EVs (500 µg) or nano-EVs (500 µg) were placed onto the agar plates. Plates were incubated for 24 h at 37 °C, and the zone of inhibition was measured. The CLSI M100-S30 guideline was used for the definition of susceptibility, intermediate susceptibility, or resistance to antibiotics for each bacterium. *Streptococcus pneumoniae* (ATCC49619) was used for quality control. All experiments were repeated at least three times.

### 2.6. Proteomic Analysis

A single colony-forming unit of GBS was inoculated into 5 mL brain–heart infusion (BHI) overnight at 30 °C. Then, 50 µL overnight culture were added into 6 mL of fresh BHI in the presence or absence of 500 µg (total proteins) micro- or nano-EVs and cultured for 6 h at 37 °C. The media were then centrifuged at 10,000× *g* for 10 min and the pellet was then suspended in PBS for proteomic analysis. The protein quantification was conducted using a bicinchoninic acid (BCA) kit and then confirmed by SDS-PAGE gel with standards. Three replications with three different placental micro- or nano-EVs were performed. The proteomic analysis was performed by iTRAQ (Majorbio Bio-Pharm Techology, Co. Ltd., Shanghai, China). Data were analysed on the Majorbio Cloud Platform (www.majorbio.com, accessed on 16 August 2021). 

### 2.7. Western Blotting

The relative levels of glycerol kinase and ribosomal protein L3 in GBS that had been treated with placental micro- or nano-EVs were measured by Western blotting. Proteins from GBS treated with placental micro- or nano-EVs were extracted with protein lysis buffer (8 M CH_4_N_2_O, 1% SDS and protease inhibitor cocktail, ThermoFisher, Shanghai, China) and all samples (20 µg of total protein for glycerol kinase or 30 µg of total protein for ribosomal protein L3) were loaded on 10% SDS-PAGE gels and electrophoresed then transferred to PVDF membranes (Millipore, Shanghai, China). Non-specific binding was blocked by incubating membranes in 5% skimmed milk in PBST for 1 h at room temperature and then membranes were incubated with rabbit polyclonal anti-glycerol kinase antibody (Abcam, ab228615, China, 1:1000) or ribosomal protein L3 antibody (ThermoFisher, Shanghai, China, 1:1000) in blocking solution for 2 h at room temperature. After washing with PBS-T three times, the membranes were incubated with goat anti-rabbit secondary antibody (1:3000) for 1 h at room temperature. After washing with PBS-T, the membranes were incubated with ECL^TM^ Prime Western blotting detection reagent (Millipore, Shanghai, China). Chemiluminescence of the membranes was detected by Tanon-5200Multi (Shanghai, China).

### 2.8. Statistical Analysis

For measurement of bacterial growth and antibiotic susceptibility, data were expressed as mean and standard deviation (SD). For analysis of the statistical difference in bacterial growth, we used a mixed model approach of repeated measures with significant time by group interaction explored using Donneth’s test to compare untreated vs. micro- or nano-EV-treated samples at each time point at an overall 5% significance level. GraphPad Prism version 8.4 (GraphPad Software, La Jolla, CA, USA) was used for this analysis. For proteomic analysis, proteins with fold changes of >1.20 or <0.83 relative to control and *p*-values lower than 0.05 were considered significantly upregulated or downregulated, respectively. A Benjamini–Hochberg multiple-comparison test after Fisher’s exact test was performed for significant differences in biological functions with *p* < 0.05. 

## 3. Results

### 3.1. Placental EVs Attenuated GBS Growth In Vitro 

To investigate the effect of placental EVs on bacterial growth, Gram-positive GBS or Gram-negative *E. coli* were cultured. As shown in Figure 1A, in the presence of placental micro-EVs (green) or nano-EVs (red), the GBS cultures entered the stationary phase significantly earlier (at 3.5 h) compared to untreated GBS (4.5 h). In contrast, no such effect was observed in the growth curves of *E. coli* treated with placental micro-EVs (green) or nano-EVs (red) when compared to untreated culture (blue) (Figure 1B). 

### 3.2. Placental EVs Inhibited the Growth of GBS

To further investigate whether placental EVs exhibit any antimicrobial activities, GBS and *E. coli* were treated with placental micro-EVs and nano-EVs for 0, 3, 6, and 12 h and then grown on Columbia blood agar plates for 24 h. As shown in Figure 2A, the colony-forming unit (CFU) counts of GBS (black bar) treated with placental micro-EVs or nano-EVs for 3, 6, or 12 h were significantly lower than the untreated controls (white bar). Treatment with nano-EVs appears to have a more pronounced effect, as less than 20 CFUs were recovered from this treatment. However, there was no significant difference between the CFU counts of *E. coli* treated or not treated with placental micro-EVs or nano-EVs at any time point (Figure 2B).

### 3.3. GBS but Not E. coli Interacted with Placental EVs

To investigate the potential underlying mechanism of the inhibitory effect of placental EVs, we then investigated the physical interaction between bacteria and placental EVs by exposing GBS or *E coli* to red fluorescently labelled placental EVs. As shown in Figure 3, the red fluorescent intensity was significantly higher in GBS treated with either red labelled placental micro- or nano-EVs, compared with GBS treated with unlabelled placental micro- or nano-EVs (Figure 3A). In contrast, the red fluorescent intensity was not different between *E. coli* treated with either red labelled placental micro- or nano-EVs and unlabelled placental micro- or nano-EVs (Figure 3B). However, when an inhibitor of phagocytosis, cytochalasin D, was added into the treatment of GBS with labelled placental EVs, the red fluorescent intensity was not changed (data not shown). In addition, the attenuation of GBS growth by placental EVs was not changed in the presence of cytochalasin D (Figure 3C,D).

### 3.4. Placental EVs Do Not Affect Bacterial Susceptibility or Resistance to Antibiotics

To investigate whether placental EVs could alter the susceptibility of bacteria to antibiotics, a Kirby–Bauer test was performed. GBS treated or untreated with placental micro-EVs and nano-EVs showed similar diameters of inhibition zones in individual tests of common antibiotics. Regardless of the placental EVs, the bacteria remained susceptible to ampicillin, penicillin, linezolid, or vancomycin, and resistant to erythromycin or clindamycin (Table 1). Similarly, no change in the diameters of inhibition zones was seen in *E. coli* treated with or without placental EVs (Table 2).

We then examined whether the placental EVs have a direct antimicrobial effect on GBS or *E. coli*. As shown in Table 3, placental micro-EVs or nano-EVs did not demonstrate any direct inhibiting effect on the growth of GBS or *E. coli* compared to the controls.

### 3.5. Proteomic Analysis Revealed Placental EVs Affect the Expression of Growth-Related Proteins

A total of 432 proteins in GBS treated with placental micro-EVs showed significantly different expression levels from the untreated control. Of them, 315 proteins were upregulated, and 117 proteins were downregulated. On the other hand, in GBS treated with placental nano-EVs, a total of 194 proteins had significantly different expression profiles from the untreated control. Of them, 136 proteins were upregulated, and 58 proteins were downregulated (Appendix A).

By Gene Ontology (GO) enrichment analysis, we found that a total of 65 biological functions were significantly downregulated in GBS treated with placental micro-EVs. Of them, the top 20 biological functions are shown in Figure 4A. These biological functions included cellular components, molecular functions, and biological processes. The top enriched downregulated biological functions are associated with protein synthesis in bacteria, such as ribosomes, intracellular ribonucleoprotein complexes, and ribonucleoprotein complexes, and 16 proteins associated with ribosome function (Table 4). In addition, 16 biological functions were significantly upregulated in GBS treated with placental micro-EVs (Figure 4B). The most significantly upregulated biological functions (*n* = 12) are involved in glutathione S-transferase activity, oxidoreductase activity, and the fatty acid biosynthetic process (Table 4).

As with the data shown in Figure 2, placental nano-EVs had a more pronounced inhibitory effect, although the exact reason was uncertain. However, our proteomic analysis showed 143 proteins overlapping between placental micro-EV- and nano-EV-treated GBS. Meanwhile, another 289 proteins were only detected in placental micro-EV-treated GBS and another 51 proteins were only detected in placental nano-EV-treated GBS. Using GO analysis for functions, six biological functions were only detected in placental micro-EV-treated GBS, and four biological functions were only detected in placental nano-EVs-treated GBS (Appendix A). The difference seen in the protein profile may explain the EVs’ different effects on GBS.

A Gene Ontology (GO) enrichment analysis also revealed a total of 32 biological functions were significantly downregulated in GBS treated with placental nano-EVs. Of them, the top 20 affecting biological functions are shown in Figure 5. These biological functions include molecular functions and biological processes. One of the main downregulated proteins is involved in kinase activity, and another 12 proteins were found to be associated with kinase activity (Table 5). In addition, although we also found that a total of two biological functions were significantly upregulated in GBS treated with placental nano-EVs, the *p* value was close to 0.05 (0.04801 and 0.04959, respectively). 

To validate the protein expression profile revealed by the proteomic analysis, we selected two proteins potentially associated with bacterial growth and detected their abundance in GBS cell lysate. As shown in Figure 6, the levels of ribosomal protein L3 (Figure 6A) were reduced in GBS treated with placental micro-EVs. The levels of glycerol kinase (Figure 6B) were reduced in GBS treated with nano-EVs compared with untreated GBS, which was consistent with the findings of the proteomic analysis.

## 4. Discussion

In this study, we found that placental micro-EVs or nano-EVs attenuated the growth of a Gram-positive bacterium, Group B Streptococcus (GBS). This attenuative effect at least partially required an interaction of placental EVs with GBS. In addition, we found that there was no direct effect of placental micro-EVs or nano-EVs on bacterial susceptibility or resistance to common antibiotics in both Gram-positive GBS and Gram-negative *E. coli*. Proteomic analysis showed that most downregulated biological functions in GBS treated with placental micro-EVs have a strong association with cellular components, especially ribosomes which are important in protein synthesis, whereas most downregulated biological functions in GBS treated with placental nano-EVs are likely to be associated with phosphorylation and cell energy.

To date, the interaction between EVs and bacteria has been understudied. However, a number of recent studies reported that EVs derived from the human placenta play multiple roles in the adaptations of pregnancy. During pregnancy, a large amount of placental EVs containing many proteins, DNA, RNA, and lipids are extruded into the maternal circulation to assist maternal physiological adaptations, including immune and vascular systems. Today, it is generally believed that EV uptake into target cells can impact their physiology. Although the underlying mechanisms of these modulatory activities are still unclear, some of these effects are mediated by proteins from placental EVs [20], whilst other effects are mediated by regulatory RNAs transferred via placental EVs to maternal cells [21]. We previously reported that phagocytosis of placental EVs by maternal endothelial cells resulted in alternation of endothelial cell physiology in both physiological and pathological pregnancy [10,11,17]. We also recently reported that phagocytosis of placental EVs inhibited ovarian cancer cell growth [19]. Collectively, this evidence suggests that placental EVs may have similar functions for bacteria. In our current study, we found that interaction with placental EVs, but maybe not phagocytosis, shortened the time to enter into the stationary phase in a Gram-positive bacterium, GBS, regardless of the size of the placental EVs. In addition, placental micro-EV or nano-EV treatments also directly inhibited the growth of GBS in liquid culture. Interestingly, these biological activities induced by placental EVs were not seen in a Gram-negative bacterium, *E. coli*. Our data showed that this could be due to the fact that *E. coli* did not interact with placental EVs, possibly due to the structural difference between the cell wall of Gram-positive and -negative bacteria. In addition, in our current study, we did not see a direct antimicrobial effect by placental micro-EVs and nano-EVs on either GBS or *E. coli* through examining inhibition zone size. Taken together, our data suggest that (1) there may be no antimicrobial compounds in placental micro-EVs or nano-EVs and (2) the effect of placental EVs on attenuating the growth of GBS could be due to the physiological changes in GBS after interaction with placental EVs.

Studies, including our own, reported that placental EVs derived from a healthy placenta carry pro-apoptotic proteins, such as TRAIL and Fas-ligand, cytokines, eicosanoids, growth factors, and their receptors [24,25,26,27], and proteins which control cell growth [20]. EVs can enter target cells with their biological activities and consequently impact the function of target cells, as reported in the field of cancer and ageing [28,29,30]. Although how the target cells respond to placental EVs is still unclear, a study reported that placental nano-EVs containing C19MC miRNA could attenuate biological replication in target cells via autophagy-mediated pathways [28]. In our current study, we found that the most significantly downregulated components are the ribosome, intracellular ribonucleoprotein complex, and ribonucleoprotein complex in GBS treated with placental micro-EVs, when compared to untreated bacteria. All these components are associated with protein synthesis in bacteria [31]. In addition, one pathway that is significantly involved in ribosome function was found by KEGG analysis in GBS treated with placental micro-EVs. The cellular ribosome contents are positively associated with the exponential phase of bacterial growth [32] and the inhibition of ribosome content function has become one of the main antibiotic targets in bacteria [33]. In our current study, we also confirmed the reduced levels of ribosomal protein L3 in GBS that had been treated with placental micro-EVs. Therefore, in our current study, we reported that the exponential phase of GBS treated with placental micro-EVs was attenuated, which could be due to inhibition of protein synthesis resulting from the downregulation of ribosome components. In addition, we also found that the most significantly upregulated biological functions in GBS with placental micro-EV treatment are involved in glutathione S-transferase activity, oxidoreductase activity, and fatty acid biosynthetic process. Glutathione S-transferase genes have also been found in bacterial operons and gene clusters involved in the degradation of aromatic compounds [34]. 

In our current study, we found that the most downregulated proteins in GBS treated with placental nano-EVs are associated with kinase activities. These kinases include the phosphotransferase system (PTS), glycerol kinase, carbamate kinase, thymidine and thymidylate kinase, adenylate kinase, and sensor histidine kinase. The PTS is responsible for the detection, transmembrane transport, and phosphorylation of its numerous sugar substrates in both Gram-negative and Gram-positive prokaryotes [35]. Glycerol kinase functions as the key enzyme of glycerol uptake and metabolism in bacteria, and carbamate kinase functions by making ATP from ADP [36]. Thymidine and thymidylate kinases are involved in bacterial DNA biosynthesis [37]. In our current study, we also confirmed the reduced levels of glycerol kinase in GBS that had been treated with placental nano-EVs. Taken together, the downregulation of kinase activities seen in GBS treated with placental nano-EVs suggested that the affected phosphorylation and cell energy could contribute to the growth attenuation of GBS. It is now well recognised that placental micro-EVs may have different functions to placental nano-EVs, due to the different cargos they carry [20], although there is overlap between the two types of placental EVs. Our study further confirmed that the underlying mechanism of the attenuation the growth of GBS is different between two types of placental EVs. 

In this study, we also found that the expression profiles of a subset of 143 GBS proteins were altered by both placental micro-EV and nano-EV treatments. Of them, there are 54 proteins with statistical significance (36 proteins were upregulated, and 18 proteins were downregulated), compared to the control (data not shown). As the statistical significance (*p* < 0.005) of upregulated proteins in GBS treated with placental micro-EVs or nano-EVs was very close to 0.05, we then only analysed the difference in downregulated proteins between GBS treated with placental micro-EVs and nano-EVs. We found that the most downregulated biological functions in GBS treated with placental micro-EVs are cellular components, molecular functions, and biological process. However, the most downregulated biological functions in GBS treated with placental nano-EVs are molecular functions and biological process. These data suggest that the underlying mechanisms that contribute to the growth attenuation may be different in the placental micro-EV and nano-EV treatments. We acknowledge the limitations of this study. Only two proteins from the protein profile altered by placental EVs were validated. Since it is more likely that a combination of multiple proteins contribute to the inhibitory effect on GBS growth, further validation should be performed.

## 5. Conclusions

In this study, we demonstrate that placental EVs attenuated the growth of Gram-positive GBS, probably through the interaction mechanism. The downregulation of cellular components and proteins associated with phosphorylation and cell energy in GBS treated with placental EVs may contribute to this effect. In addition, placental EVs did not impact susceptibility or resistance of GBS to antibiotics, further supporting the hypothesis that the attenuative effect was due to physiological changes in GBS by the treatments.

## Figures and Tables

**Figure 1 biology-10-00664-f001:**
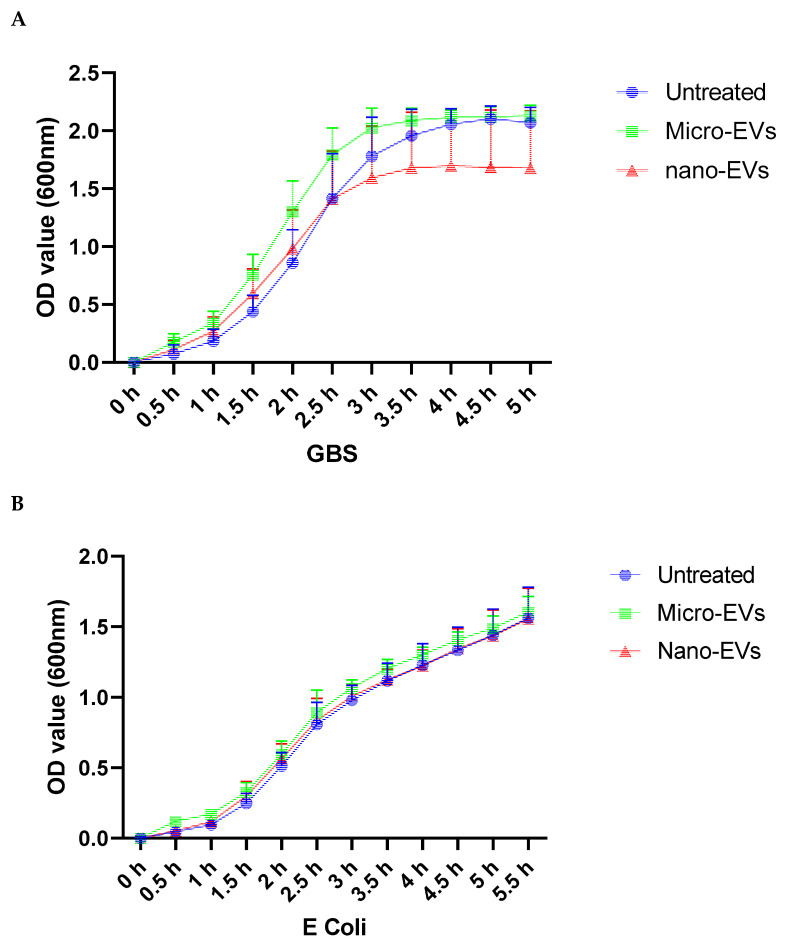
Growth curves of GBS (**A**), but not *E. coli* (**B**) in the presence of either placental micro- or nano-EV treatment were significantly attenuated, compared to untreated GBS (*p* < 0.0001).

**Figure 2 biology-10-00664-f002:**
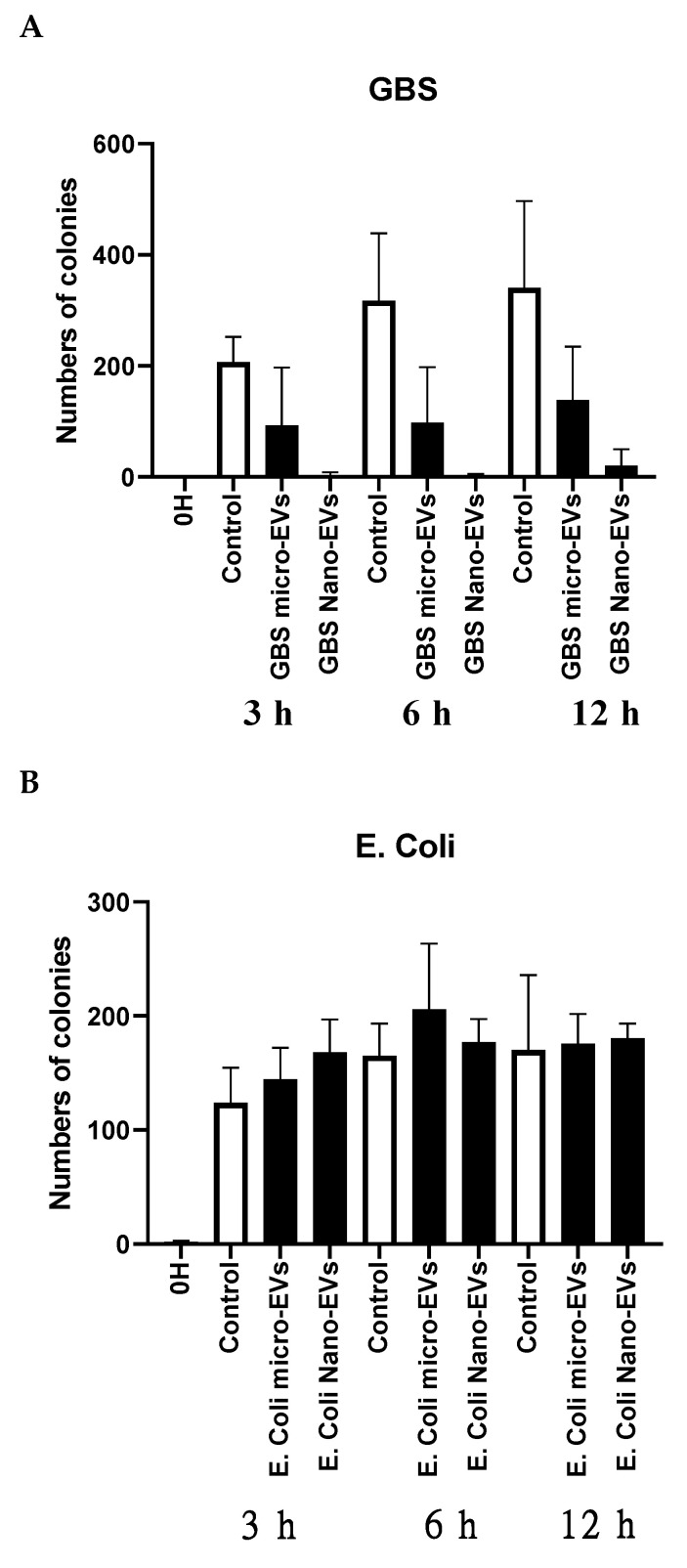
The colony-forming unit (CFU) counts of GBS (**A**), but not *E. coli* (**B**) after treatment with placental micro-EVs or nano-EVs (black bars) were inhibited, compared to untreated GBS (white bars).

**Figure 3 biology-10-00664-f003:**
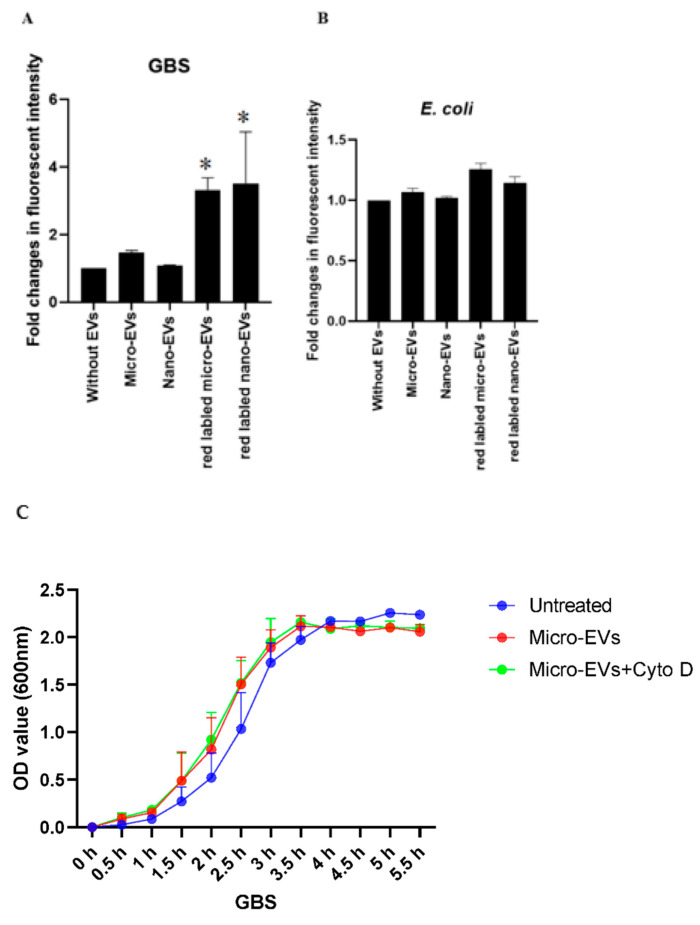
Fluorescent intensity was significantly higher in GBS treated with red labelled placental micro- or nano-EVs (**A**), compared to GBS treated with unlabelled placental EVs (* *p* = 0.0159). However, there was no difference in the fluorescent intensity in *E. coli* treated with red labelled placental micro- or nano-EVs (**B**, *p* > 0.05, ANONA). The attenuating effect of GBS growth by placental micro-EVs (**C**) or nano-EVs (**D**) was not blocked in the presence of cytochalasin D (10 µM).

**Figure 4 biology-10-00664-f004:**
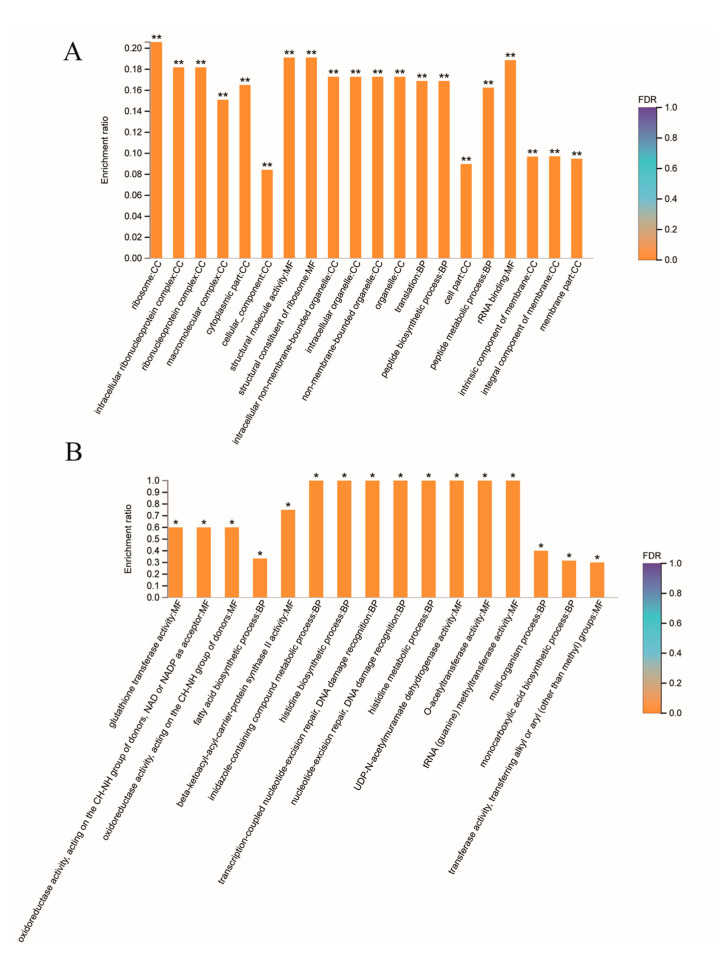
Gene Ontology analysis of the top 20 downregulated biological functions (**A**) or top 16 upregulated biological functions (**B**) in GBS treated with placental micro-EVs, compared with the untreated controls. (BH multiple-comparison test after Fisher’s exact test. ** FDR < 0.01, * FDR < 0.05).

**Figure 5 biology-10-00664-f005:**
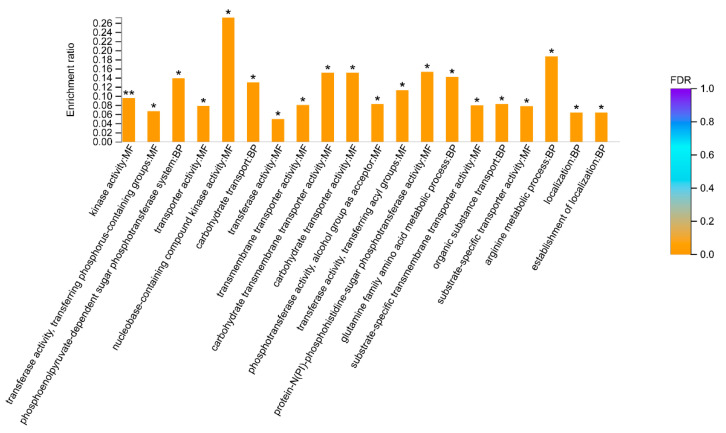
Gene Ontology analysis of the top 20 biological functions downregulated in GBS treated with placental nano-EVs, compared with the untreated controls. (BH multiple-comparison test after Fisher’s exact test. ** FDR < 0.01, * FDR < 0.05).

**Figure 6 biology-10-00664-f006:**
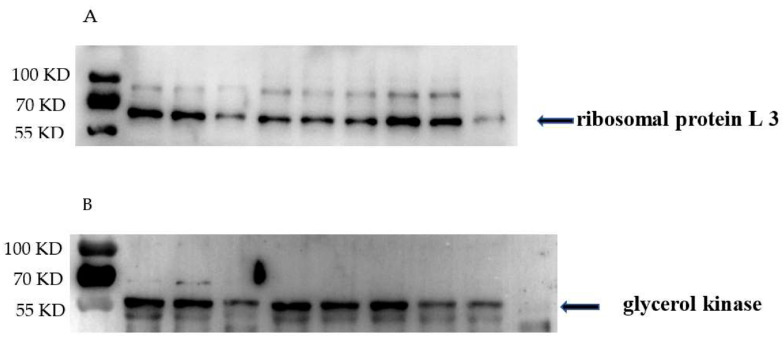
Western blots demonstrating that (**A**) the levels of ribosomal proteins L3 were significantly reduced in GBS treated with placental micro-EVs and (**B**) the levels of glycerol kinase in GBS treated with placental nano-EVs were significantly reduced, compared with untreated GBS (*n* = 3).

**Table 1 biology-10-00664-t001:** Diameter of antibiotic inhibition zone (mm) measured for GBS treated with or without placental EVs.

Name of Antibiotics	Without EVs	With Micro-EVs	With Nano-EVs
Susceptible (Standard)			
Ampicillin (10 µg) (≥24)	29.3	28.3	30
Penicillin (15 units) (≥24)	31.3	31.3	29.3
Linezolid (30 µg) (≥21)	30	31	29
Vancomycin (30 µg) (≥17)	21	21	20
Resistant (Standard)			
Erythromycin (15 µg) (≤15)	8	8	7.7
Clindamycin (2 µg) (≤15)	6	6	8

**Table 2 biology-10-00664-t002:** Diameter of antibiotic inhibition zone (mm) measured for *E. coli* treated with or without placental EVs.

Name of Antibiotics	Without EVs	With Micro-EVs	With Nano-EVs
Susceptible (Standard)			
Fosfomycin (200 µg) (≥16)	27	27	27
Piperacillin/Tazobactam (100/10 µg) (≥21)	29	26	27
Meropenem (10 µg) (≥23)	30	29	29
Ceftazidime (30 µg) (≥21)	26.5	26.7	26.7
Ciprofloxacin (5 µg) (≥26)	31.2	30.7	30.7
Amikacin (30 µg) (≥17)	22.4	22.4	22
Resistant (Standard)			
Ampicillin (10 µg) (≤13)	6	6	6
Piperacillin (100 µg) (≤17)	10	10	13
Ampicillin/Sulbactam (10/10 µg) (≤11)	6	7	8.5
Ceftriaxone (30 µg) (≤19)	7.25	7.5	6.15

**Table 3 biology-10-00664-t003:** The diameter of inhibition zone (mm) by treatment with placental EVs.

	*E. coli*	GBS
Without placental EVs	6	6
With placental micro-EVs	6	6
With placental nano-EVs	6	6

**Table 4 biology-10-00664-t004:** Differentially downregulated or upregulated proteins in GBS treated with placental micro-EVs involved in the ribosomal components, glutathione transferase activity, oxidoreductase activity, and fatty acid biosynthetic process.

Protein Accession	Regulation	Functional Description
Ribosome		
A0A1C0BEJ0	down	50S ribosomal protein L21, rRNA binding
A0A4U3JET0	down	30S ribosomal protein S1, nucleic acid binding
A0A6A4TZH8	down	50S ribosomal protein L3, rRNA binding
Q3K0C9	down	50S ribosomal protein L35, structural constituent of ribosome
Q8DYG1	down	50S ribosomal protein L11, large ribosomal subunit rRNA binding
Q8DZZ2	down	30S ribosomal protein S20, rRNA binding
R4Z934	down	30S ribosomal protein S11, rRNA binding
V6YZ47	down	50S ribosomal protein L3, rRNA binding
V6YZ80	down	30S ribosomal protein S13, RNA/tRNA binding
V6Z128	down	30S ribosomal protein S9, structural constituent of ribosome
V6Z135	down	30S ribosomal protein S11, rRNA/mRNA 5′-UTR/small ribosomal subunit rRNA binding
V6Z275	down	S1 RNA-binding protein, nucleic acid binding
V6Z2C6	down	50S ribosomal protein L7/L12, structural constituent of ribosome
V6Z6Z4	down	Uncharacterised protein, nucleic acid binding
Q8E3E6	down	30S ribosomal protein S7, rRNA binding
V6Z0L9	down	50S ribosomal protein L10, large ribosomal subunit rRNA binding
Glutathione Transferase Activity		
A0A0G2Z569	up	Glutathione S-transferase, omega
S8FQZ0	up	GST C-terminal domain-containing protein
V6Z179	up	S-transferase
Oxidoreductase activity, acting on the CH-NH group of donors, NAD or NADP as acceptor		
A0A0E1EHT5	up	Nitroreductase
Q8E7I3	up	Pyrroline-5-carboxylate reductase
S9AZA4	up	Bifunctional protein FolD
Fatty Acid Biosynthetic Process		
A0A0G9JEY8	up	3-oxoacyl-[acyl-carrier-protein] synthase 2
A0A656G1D0	up	4′-phosphopantetheinyl transferase
Q8E5S2	up	Uncharacterised protein
S8FH53	up	Biotin carboxyl carrier protein of acetyl-CoA carboxylase
S9B2D4	up	3-oxoacyl-[acyl-carrier-protein] synthase 2
V6Z187	up	Biotin carboxylase

**Table 5 biology-10-00664-t005:** Downregulated proteins in GBS treated with placental nano-EVs involved in the kinase activity.

Protein Accession	Regulation	Functional Description
A0A0H1L4V8	down	PTS beta-glucoside transporter subunit IIBCA
A0A0H1NG81	down	Putative galactitol operon regulator (Transcriptional antiterminator), BglG family PTS system, mannitolfructose-specific IIA component
A0A0H1U8Y4	down	Glycerol kinase
A0A380IIL8	down	Carbamate kinase
A0A380IY32	down	PTS system, sucrose-specific IIB component/PTS system, sucrose-specific IIC component/PTS system, sucrose-specific IIA component
A0A5N0LHT3	down	Thymidine kinase
A0A656G179	down	PTS system sucrose-specific IIBC component
P65204	down	Adenylate kinase
Q8DX04	down	Sensor histidine kinase
Q8E2P0	down	Uncharacterised protein
V6Z3K8	down	Thymidylate kinase
V6Z7B9	down	Lipid kinase

## Data Availability

The datasets used and/or analysed during the current study are available from the corresponding author on reasonable request.

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
