# Peer review of "Downregulation of Ribosomal Contents and Kinase Activities Is Associated with the Inhibitive Effect on the Growth of Group B Streptococcus Induced by Placental Extracellular Vesicles"

_biology, 2021, doi:10.3390/biology10070664_

Round 1

Reviewer 1 Report

In their paper titled " Down-regulation of ribosomal contents and kinase activities is associated with the inhibitive effect on the growth of Group B Streptococcus induced by placental extracellular vesicles", Gao et al. describes that the important role of placental EV on the function of bacteria such as GBS and E Coli. The authors assessed that placental EV could reduce the physiological functions of GBS. This is a very interesting manuscript and valuable for many researchers. Also, the aim is sufficiently impressive. I believe that this manuscript is suitable for publication in this journal.

Minor concern:

Isn't the figure legend in figure6 slightly wrong?

Author Response

Dear Reviewer

many thanks for your positive comments and support. 

Minor concern:

Isn't the figure legend in figure6 slightly wrong?

Answer: Sorry for the mistakes in the figure legend for figure 6. We have now corrected the error.  

Reviewer 2 Report

In this manuscript, the authors demonstrated that the placental produced extracellular vesicles (EVs) could potentially inhibit the growth of Group B Streptococcus (GBS). By doing proteomics analysis, the authors claimed that the inhibitory effect is likely associated with some bacterial pathways. The manuscript was organized and prepared well. The concept itself is interesting and the findings may benefit the field. However, the current story has couple of issues which have to be addressed before consideration for publication.

  1. The background on the effect/interaction of EVs and bacterial cells need to be extensively introduced/discussed. The function/effect of EVs on mammalian cells has noting to do with this story, no need to emphasize.
  2. The inhibition effect of EVs on GBS growth is small. The error bars and statistic analysis of Figure 1 are missing. From the curve (Figure 1A), it is hard to draw a conclusion that “EVs attenuated GBS growth”. This issue is directly associated with the biggest selling point of this story. More detailed assays are required to confirm the inhibitory effect.
  3. It is good to know the specificity on the EV side – it is placental specific or the EVs generated by other tissues also have the same effect?
  4. The quantification method of proteomic analysis is missing. Also, the centrifuge at 10,000 g to harvest bacterial cells is too high, some of the EV proteins may also go to the pellet and interfere the proteomic analysis.
  5. The A and B of Supplement Figure 1 are inverted, A should be nono-EVs, B should be micro-EVs.
  6. Why the nano-EVs showed the more “pronounced” inhibitory effect in Figure 2A but less dramatic changes on proteomic level? Also, it is good to see more analyses on the comparison of micro-EV and nono-EV treated proteome profiles to illustrate the common and/or unique pathways.
  7. The proteomic part is not conclusive and not fully validated. You got these ribosomal proteins and kinases might only due to they are abundant. Figure 6 is the only one piece of “validation”, which has big variation, lacks proper loading control, and has no statistical values. So far, I cannot tell those significantly regulated pathways are real or not.

Author Response

1. The background on the effect/interaction of EVs and bacterial cells need to be extensively introduced/discussed. The function/effect of EVs on mammalian cells has noting to do with this story, no need to emphasize.

Answer: To date, there is little information on the interaction of EVs and bacteria. The aims of our study were to investigate the function of placental EVs on bacterial cells. This is because recently many studies have reported the functions of placental EVs on human pregnancy and placental EVs contain many cell proliferation or cell death-associated proteins. This is the background for performing this study. The function of placental EVs on bacterial cells has not been previously investigated. We have now modified the introduction and discussion section (lines 57-70, and line 83-87).

2.The inhibition effect of EVs on GBS growth is small. The error bars and statistic analysis of Figure 1 are missing. From the curve (Figure 1A), it is hard to draw a conclusion that “EVs attenuated GBS growth”. This issue is directly associated with the biggest selling point of this story. More detailed assays are required to confirm the inhibitory effect.

Answer: we have now added the statistical analysis for GBS and E Coli growth curve and (line 183-187). In this study, we also enumerated the colony-forming units as a second method to confirm our hypothesis. In figure 1, we found that the time to reaching stationary phase is significantly earlier in placental EV treated GBS than untreated GBS (p<00001).

3. It is good to know the specificity on the EV side – it is placental specific or the EVs generated by other tissues also have the same effect?

Answer: This is a great point. We did not test this effect by EVs from other sources in this study, however, our previous study reported that the inhibition of ovarian cancer cell growth is specific to placental EVs, not other EVs (ref 19).

4. The quantification method of proteomic analysis is missing. Also, the centrifuge at 10,000 g to harvest bacterial cells is too high, some of the EV proteins may also go to the pellet and interfere the proteomic analysis.

Answer: We have now added the method for protein quantification (line 159-161). We do agree that the centrifugation at 10,000 g was high. However, the collection of placental micro-EVs needs to be centrifugated at 20,000 g for 1 hour, or collection of placental nano-EVs needs to be centrifugated at 100,000 g for 1 hour. This suggested that the contamination of placental EVs or bacterial EVs in the pellet of GBS is unlikely.  

5. The A and B of Supplement Figure 1 are inverted, A should be nono-EVs, B should be micro-EVs.

Answer: Sorry for the mistake. We have now collected the error.

6. Why the nano-EVs showed the more “pronounced” inhibitory effect in Figure 2A but less dramatic changes on proteomic level? Also, it is good to see more analyses on the comparison of micro-EV and nono-EV treated proteome profiles to illustrate the common and/or unique pathways.

Answer: it is well-known that micro-EVs and nano-EVs carry different cargos. It is also not surprising that the effect on attenuation of GBS growth is different. Our previous study (ref 19 in the manuscript) showed the different effects on ovarian cancer growth by placental macro, micro, and nano-EVs.      

We have now also analysed the difference in the protein profile in GBS between micro-EVs treated and nano-EVs treated. There were 143 proteins overlapping between micro-EVs and nano-EVs treated GBS. While other 289 proteins were only detected in micro-EVs treated GBS and other 51 proteins were only detected in nano-EVs treated GBS. Using GO analysis for function, 6 proteins were only detected in micro-EVs treated GBS, and 4 proteins were only detected in nano-EVs treated GBS. The difference in the protein profile explains the different effect on GBS. We now included this data in the supplementary file (Supplement Figure 2-5).   

7. The proteomic part is not conclusive and not fully validated. You got these ribosomal proteins and kinases might only due to they are abundant. Figure 6 is the only one piece of “validation”, which has big variation, lacks proper loading control, and has no statistical values. So far, I cannot tell those significantly regulated pathways are real or not.

Answer: As pointed by reviewer 3 as well, we do agree that validation with 2 proteins was not fully justified. The main aim of this study is to investigate the effect of placental EVs on GBS growth. Since it is more likely that a combination of multiple proteins contributes to the inhibitory effect on GBS growth, further validation should be performed. We have acknowledged this limitation in the discussion section (lines 439-442).   

For western blotting, we loaded 20 ug proteins of each sample for glycerol Kinase or 30 ug proteins of each sample for ribosomal protein L3 (see line 172). To date, the loading control for GBS is limited, we have tried to use GAPDH as a loading control. But the levels of GAPDH were very low in GBS.

Reviewer 3 Report

The current study entitled "Down-regulation of ribosomal contents and kinase activities is associated with the inhibitive effect on the growth of Group B Streptococcus induced by placental extracellular vesicles" investigates the inhibitory effect of placental EVs on the growth of GBS. 

This is an interesting study planned and performed well, however, needs to address few points for better presentation-

1- As shown in the result, one of the primary steps of EV mediated effects on GBS growth and survival is its interaction with EVs (phagocytosis) so data regarding phagocytosis inhibitor is important and should be shown if available.

2- It's good that the authors validated the protein expression profiles by western blotting but the selection of just 2 proteins is not fully justified. Additionally, western blot data is not matching the result text. Protein L3 bands are unchanged in the Nano-EV treated group but the text says it's decreased. 

3- How western blotting results were normalized? There should be some control/housekeeping genes to demonstrate equal loading and densitometric data will be valuable. 

Author Response

Dear Reviewer

many thanks for your time and comments on our manuscript. 

1- As shown in the result, one of the primary steps of EV mediated effects on GBS growth and survival is its interaction with EVs (phagocytosis) so data regarding phagocytosis inhibitor is important and should be shown if available.

Answer: We used cytochalasin D (an inhibitor of phagocytosis) to investigate whether phagocytosis is involved in this effect. However, our data showed that the attenuation of GBS growth by placental EVs was not blocked in the presence of cytochalasin D. We have now added this data in the revised manuscript as request (Figure 3C-3D).  

2- It's good that the authors validated the protein expression profiles by western blotting but the selection of just 2 proteins is not fully justified. Additionally, western blot data is not matching the result text. Protein L3 bands are unchanged in the Nano-EV treated group but the text says it's decreased. 

Answer: We do agree that validation with 2 proteins was not fully justified. The main aim of this study is to primarily investigate the effect of placental EVs on GBS growth. The two proteins we selected were abundant and are involved in cell energy and protein synthesis. We have added this limitation in the discussion section.  

We apologize for the wording in the validation section being confusing. We have now re-worded these sentences including the figure legend for Figure 6 as suggested by reviewer 1 (lines 341-344).  

3- How western blotting results were normalized? There should be some control/housekeeping genes to demonstrate equal loading and densitometric data will be valuable. 

Glycerol kinase

GAPDH

   Untreated      Micro-EVs    Nano-EVs group

Answer: For western blotting, we loaded 20 ug proteins of each sample for glycerol Kinase or 30 ug proteins of each sample for ribosomal protein L3 (see line 172). To date, the loading control for GBS is limited, we have tried to use GAPDH as a loading control. But the levels of GAPDH were very low in GBS (see below as an example).

Round 2

Reviewer 2 Report

The authors have answered all my concerns in the revised version. Good job.

Author Response

Dear reviewer

Many thanks for your efforts and time in reviewing our manuscript. 

Kind Regards

Qi